# *Chlorella vulgaris*-Derived Biochars for Metribuzin Removal: Influence of Thermal Processing Pathways on Sorption Properties

**DOI:** 10.3390/ma18143374

**Published:** 2025-07-18

**Authors:** Margita Ščasná, Alexandra Kucmanová, Maroš Sirotiak, Lenka Blinová, Maroš Soldán, Jan Hajzler, Libor Ďuriška, Marián Palcut

**Affiliations:** 1Institute of Integrated Safety, Faculty of Materials Science and Technology, Slovak University of Technology, J. Bottu 25, 917 24 Trnava, Slovakia; margita.scasna@stuba.sk (M.Š.); alexandra.kucmanova@stuba.sk (A.K.); maros.sirotiak@stuba.sk (M.S.); lenka.blinova@stuba.sk (L.B.); maros.soldan@stuba.sk (M.S.); 2Materials Research Center, Faculty of Chemistry, Brno University of Technology, Purkyňova 464/118, 612 00 Brno, Czech Republic; jan.hajzler@vut.cz; 3Institute of Materials Science, Faculty of Materials Science and Technology, Slovak University of Technology, J. Bottu 25, 917 24 Trnava, Slovakia; libor.duriska@stuba.sk

**Keywords:** biochar, carbonaceous materials, hydrothermal carbonization, metribuzin adsorption, microalgae, pesticide removal, pyrolysis

## Abstract

Carbonaceous sorbents were prepared from *Chlorella vulgaris* via hydrothermal carbonization (200 °C and 250 °C) and slow pyrolysis (300–500 °C) to assess their effectiveness in removing the herbicide metribuzin from water. The biomass was cultivated under controlled laboratory conditions, allowing for consistent feedstock quality and traceability throughout processing. Using a single microalgal feedstock for both thermal methods enabled a direct comparison of hydrochar and pyrochar properties and performance, eliminating variability associated with different feedstocks and allowing for a clearer assessment of the influence of thermal conversion pathways. While previous studies have examined algae-derived biochars for heavy metal adsorption, comprehensive comparisons targeting organic micropollutants, such as metribuzin, remain scarce. Moreover, few works have combined kinetic and isotherm modeling to evaluate the underlying adsorption mechanisms of both hydrochars and pyrochars produced from the same algal biomass. Therefore, the materials investigated in the present work were characterized using a combination of standard physicochemical and structural techniques (FTIR, SEM, BET, pH, ash content, and TOC). The kinetics of sorption were also studied. The results show better agreement with the pseudo-second-order model, consistent with chemisorption, except for the hydrochar produced at 250 °C, where physisorption provided a more accurate fit. Freundlich isotherms better described the equilibrium data, indicating heterogeneous adsorption. The hydrochar obtained at 200 °C reached the highest adsorption capacity, attributed to its intact cell structure and abundance of surface functional groups. The pyrochar produced at 500 °C exhibited the highest surface area (44.3 m^2^/g) but a lower affinity for metribuzin due to the loss of polar functionalities during pyrolysis. This study presents a novel use of *Chlorella vulgaris*-derived carbon materials for metribuzin removal without chemical activation, which offers practical benefits, including simplified production, lower costs, and reduced chemical waste. The findings contribute to expanding the applicability of algae-based sorbents in water treatments, particularly where low-cost, energy-efficient materials are needed. This approach also supports the integration of carbon sequestration and wastewater remediation within a circular resource framework.

## 1. Introduction

The negative consequences of contaminating the aquatic environment with pesticides represent a growing environmental concern due to their persistence and bioaccumulative properties. Pesticides contaminate water sources, threaten aquatic organisms, and reduce water quality through runoff into rivers, lakes, and groundwater contamination [1]. Their widespread use and non-targeted effects harm other species, such as pollinators; disrupt the ecological balance; and threaten agricultural production. The persistence of these compounds in the environment leads to bioaccumulation and biomagnification in the food chain [2], which is particularly detrimental to top predators. In addition to their environmental effects, pesticides pose serious health risks to humans, as both short-term and long-term exposure can lead to cancer and neurological disorders, which can be particularly harmful to vulnerable groups, such as children, pregnant women, and immunocompromised individuals [3,4].

Metribuzin, a triazinone herbicide, is commonly used for controlling dicotyledonous plants. It is a frequent and moderately persistent contaminant in surface and groundwater systems. In agricultural catchments, metribuzin has been detected in surface waters at concentrations typically below 2 μg/L, with local peaks occurring following application or rainfall events [5,6,7,8,9]. As such, it can adversely affect aquatic ecosystems and threaten human health [10]. The water solubility of metribuzin can be as high as 1200 mg/L, thereby significantly increasing its transport capacity in the soil–water system [11]. The half-life of metribuzin in the soil can exceed 30 days. Webb et al. [12] report a half-life of 145 days, while Lechón et al. and Di et al. [13,14] report half-lives in the range of 11 to 46 days. Its chemical stability can range from several weeks to months, depending on the soil type, temperature, and pH [15]. Despite the availability of several water treatment technologies, the effective removal of metribuzin and other contaminants remain challenging, as it relies on conventional physicochemical methods that are often either costly or generate secondary waste [16]. Recent reviews highlight these limitations and suggest that microalgae-based or carbonaceous sorbents may provide more sustainable alternatives [17,18].

Over the past few decades, the adsorption of pollutants onto suitable sorbents has garnered considerable attention as a low-cost and sustainable alternative to more conventional methods. Of the many sorbents available, biochar is the most promising due to its favorable surface chemistry, tunable porosity, low production cost, and the possibility of utilizing diverse waste biomass as feedstock [19]. Biochar is a carbonaceous material that is prepared by thermal conversion of biomass in the absence of oxygen [20]. It can be used as an adsorbent for contaminants in both soil and water, effectively demonstrating its practical application in decontamination [16].

However, many studies focus on a single type of biochar, typically derived from lignocellulosic biomass and produced under fixed pyrolysis conditions, limiting the broader understanding of how process parameters influence sorption performance [21,22]. In contrast, the present study systematically compares the sorption efficiency of hydrochars and pyrochars derived from the same microalgal biomass (*Chlorella vulgaris*) but produced under different hydrothermal and pyrolytic conditions. This approach enables a more precise evaluation of how temperature and reaction time affect the physicochemical properties of the sorbents and their subsequent interaction with metribuzin. Such comparative studies remain scarce, particularly for sorbents derived from algal feedstocks, which offer distinct advantages in terms of their uniform composition and high nitrogen content [17,18].

Biochar derived from microalgal biomass exhibits several favorable sorption properties, such as a high specific surface area, abundant oxygen- and nitrogen-containing functional groups, and heteroatom doping (e.g., N, P, and S) due to the intrinsic composition of algal feedstocks [23,24]. Carbonaceous materials obtained from *Chlorella vulgaris* exhibit distinct morphological and chemical surface features, including an irregular porous structure and the presence of nitrogen moieties and functional groups, which support efficient interactions with a broad range of organic and inorganic contaminants [22,25].

The strong affinity of algal biochars toward polar organic compounds, including pesticides, is attributed to π–π interactions, electrostatic forces, and hydrogen bonding, which enhance their applicability in water purification processes [21,26,27,28].

*Chlorella vulgaris* is a unicellular green alga recognized for its rapid biomass accumulation and resilience to diverse growth conditions. It can be cultivated in wastewater-rich environments [29], where it simultaneously contributes to nutrient removal and generates biomass suitable for further valorization [17,18]. The dual role of *Chlorella vulgaris*, first as a biosorbent and then as a precursor for biochar production, underscores the integrative character of this approach, which is consistent with circular bioeconomy and biorefinery strategies [25,30,31,32,33,34].

Thermal treatment of microalgal biomass yields carbonaceous solids through methods such as slow pyrolysis, hydrothermal carbonization (HTC), or torrefaction [24,35]. Each process yields materials with distinct physicochemical characteristics, depending on parameters such as temperature, reaction time, and atmosphere [22]. Pyrochars typically possess a more developed porous structure and higher aromaticity, while hydrochars retain a greater number of oxygenated and nitrogenous surface groups. While both types of chars have been individually investigated for pollutant removal, there is a notable lack of studies directly comparing the sorption performance of pyrochars and hydrochars derived from the same microalgal feedstock under varied conditions. This study deals with this issue through a systematic evaluation of the removal of metribuzin using *Chlorella vulgaris*-based sorbents produced by both slow pyrolysis and HTC at different temperatures [21,26].

This study investigates the feasibility of the removal of metribuzin from water using biochars produced from *Chlorella vulgaris* via two thermal conversion pathways—slow pyrolysis and hydrothermal carbonization (HTC)—each applied at multiple temperatures. The *Chlorella vulgaris* biomass was cultivated under controlled laboratory conditions (light, temperature, and nutrient availability) to ensure a uniform composition and reproducible quality. By applying different thermal treatments to the same feedstock, the resulting sorbents could be directly compared without interference from feedstock variability.

To characterize the materials, we determined their pH, ash content, yield, and oxidizable organic carbon content. We also examined the surface chemistry and morphology using FTIR and SEM, and in the case of pyrochars, we also measured the specific surface area. The sorption performance of each biochar was evaluated using kinetic and isotherm modeling to better understand the means of their interaction with metribuzin.

To our knowledge, this is the first study to systematically compare the feasibility of using hydrochars and pyrochars derived from *Chlorella vulgaris* under multiple production temperatures for the specific removal of metribuzin. While previous research has explored the use of biochars derived from various algal sources, few studies have investigated sorbents produced specifically from *Chlorella vulgaris* in the context of organic micropollutant removal and even fewer have applied kinetic and isotherm modeling to elucidate the adsorption mechanisms. Biochars derived from *Chlorella vulgaris* retain nitrogenous surface groups that may interact favorably with metribuzin molecules, supporting sorption through mechanisms such as π–π interactions, hydrogen bonding, and electrostatic forces. We hypothesize that hydrochars produced at lower HTC temperatures retain a higher fraction of polar surface groups and a more preserved cell structure, resulting in stronger interactions with metribuzin despite limited porosity. In contrast, pyrochars formed at higher temperatures are expected to have a greater surface area but fewer functional groups. By focusing on a single, well-characterized microalgal feedstock cultivated under controlled conditions, this study provides a targeted understanding of how thermal processing pathways influence the adsorption behavior of *Chlorella vulgaris*-based sorbents toward organic micropollutants, such as metribuzin, which contain nitrogen functionalities likely to interact with nitrogen-rich surface groups.

## 2. Materials and Methods

All chemicals used in this study were of analytical grade and were obtained from MERCK spol. s r. o. (a subsidiary of Merck KGaA, Darmstadt, Germany) and FISHER Slovakia, spol. s r. o. (a subsidiary of Thermo Fisher Scientific Inc., Waltham, MA, USA).

### 2.1. Microalgae Cultivation and Biomass Preparation

A sterile culture of *Chlorella vulgaris* was obtained from the Culture Collection of Algae and Lichens (CCALA), which is a member of the World Federation for Culture Collections (WFCC) and the World Data Centre for Microorganisms (WDCM). The cultivation was carried out mixotrophically in closed 1000 mL Erlenmeyer flasks in a sterile liquid standard BG11 medium [36]. During cultivation, the microbial biomass was continuously stirred with magnetic stirrers and aerated using air pumps (ALITA Air Pump AL-6SA, ALITA Industries, Baldwin Park, CA, USA). The aeration of the biomass was performed using clean air filtered through a 0.22 μm Polytetrafluoroethylene (PTFE) filter. Carbon dioxide (CO_2_) from the air enabled faster growth, allowing cells to reach the exponential phase of growth earlier. The continuous mixing and aeration of the microbial biomass prevented algae from settling on the walls of the photobioreactors, ensuring better light transmission. During cultivation, the biomass was illuminated at 2000 lx, with a light/dark cycle of 16/8 h at laboratory temperature (25 ± 1 °C).

The cultivation of microalgae in closed photobioreactors offers several advantages, including higher biomass productivity, a controlled microenvironment, and a reduced risk of contamination. Key parameters such as temperature and pH are also strictly controlled to achieve maximum growth [32].

The biomass gain was monitored by regular measurements of optical density until the exponential phase of growth was reached. Optical density was measured at a wavelength of 680 nm using a UV/VIS spectrophotometer (Agilent Cary 60 UV-Vis, Santa Clara, CA, USA). A wavelength of 680 nm was chosen, as this corresponds to the absorption maximum of chlorophyll a in *Chlorella vulgaris*, which is a direct indicator of algal biomass concentration. While π-π* transitions of organic compounds typically appear in the 300–400 nm region, these are not specific to algal biomass and may be influenced by other organic substances present in the medium. In contrast, absorbance at 680 nm is highly specific to the photosynthetic pigments of microalgae and is widely used in the literature for accurate monitoring of algal growth [24,37,38,39,40].

The harvesting of microbial biomass was performed through sedimentation. The biomass was rinsed three times with distilled water and then dried in an oven (Memmert UNB 500, EMIN GROUP, Hackensack, NJ, USA) at 105 °C to a constant weight, as verified by repeated weighing at regular intervals. Once the mass stabilized, the samples were transferred to a desiccator to prevent moisture uptake.

### 2.2. Hydrothermal Carbonization

Hydrothermal carbonization was conducted in a laboratory autoclave reactor (model PPL100, Tefic Biotech, Xi’an, China), constructed from 304 stainless steel and featuring an internal polyphenylene (PPL) liner. A PPL liner provides enhanced thermal and chemical resistance, making the reactor suitable for high-temperature aqueous processing. The PPL liner was filled with 6.00 g of biomass and 60 mL of distilled water, which were then stirred for 5 min. After sealing, the shell was placed in a steel reactor and inserted into a muffle furnace. The heating rate, up to the desired temperature (200 °C and 250 °C), was 5 ± 0.5 °C/min. The retention time was 60 min. After the reaction, the reactor was passively cooled in air to laboratory temperature (25 ± 1 °C). Once it reached this temperature, the mixture was opened and centrifuged using a Nahita 2640/12 centrifuge (ManualsLib, Shenzhen, China) at 4000 rpm for 120 s at laboratory temperature (25 ± 1 °C) to separate the solid and liquid phases. The solid particles were washed with distilled water and dried in an oven (Memmert UNB 500) at 105 °C to constant weight, confirmed by repeated weighing. Dried and ground samples were stored in a desiccator to avoid moisture absorption.

### 2.3. Slow Pyrolysis

Approximately 5.00 g of dry biomass from *Chlorella vulgaris* was weighed into 3/60 porcelain crucibles and pyrolyzed in a muffle furnace (Laboratory oven LAC LE05/11 with controller Ht60B, LAC, s.r.o., Židlochovice, Czech Republic) using slow pyrolysis at different temperatures (300, 350, 400, 450, and 500 °C) with a temperature gradient of 10 °C/min in a gaseous nitrogen environment with a flow rate of 20 mL/min and a residence time of 60 min. At the end of the reaction time, the prepared biochars were cooled to laboratory temperature (25 ± 1 °C) and subsequently ground. The air moisture was removed by drying the biochars in a drying oven (Memmert UNB 500) at 105 °C to constant weight, confirmed by repeated weighing, and then, it was stored in a desiccator until further use in the experiments.

### 2.4. Biochar Characterization and Structural Analysis

The pH was measured using a pH meter, Multi 340i, with a Sentix 41 probe. The active pH of the biochars was measured using ultrapure water with a conductivity of less than 0.2 μS/cm. Biochar samples with particle sizes smaller than 1 mm were mixed with ultrapure water at a 1:10 (*w*/*v*) ratio [41].

The exchangeable pH of the biochars was determined using a 1 mol/L KCl solution at a 1:10 (*w*/*v*) ratio. In both procedures, the mixtures were stirred with a magnetic stirrer for 60 min and then allowed to settle for 10 min before the pH was measured using a calibrated pH meter [42]. The pH values were measured in triplicate. The measured values were reproducible, with standard deviations remaining within ±0.004 across all samples.

The yield of each biochar was calculated as the ratio of the dry mass of the produced biochar to the initial dry biomass and expressed as a percentage (%). All samples were weighed using a digital analytical balance (RADWAG AS 310/C/2, readability 0.1 mg, Radom, Poland). Six independent replicates were performed for each temperature condition (200–500 °C). The results are reported as means ± standard deviations.

The determination of ash content in the biomass sample from *Chlorella vulgaris* was performed in six replicates [43]. Approximately 1.00 g of dried biomass was weighed into a lidded annealing crucible using an analytical balance. The crucible was then placed in a muffle furnace (laboratory oven LAC LE05/11 with controller Ht60B), and the temperature was gradually increased over 30–50 min to 250 °C, where it was held for 60 min. Subsequently, the temperature was raised to 550 ± 10 °C and maintained for 120 min. At the end of the heating program, the crucible was transferred to a desiccator to cool to laboratory temperature (25 ± 1 °C). After cooling, the crucible containing ash was weighed. The ash content was calculated as a weight percentage (*w*/*w*).

The oxidizable organic carbon (C_org_) content in the solid fraction of the biochars was determined spectrophotometrically using a UV/VIS spectrophotometer, Agilent Cary 60 UV-Vis at a wavelength of 590 nm, following the Turin method as modified by Nikitin [44]. This method is based on the oxidation of organic carbon by potassium dichromate in concentrated sulfuric acid. The amount of trivalent chromium (Cr^3+^) formed due to the reduction of hexavalent chromium (Cr^6+^) is stoichiometrically equivalent to the amount of oxidizable organic carbon in the sample. Three independent replicates were performed for each biochar (prepared at 200–500 °C), as well as biomass. The results are reported as means ± standard deviations.

The samples were characterized using Fourier-transform infrared spectroscopy (FTIR). The spectra were acquired using a Varian 660 MidIR Dual MCT/DTGS Bundle FTIR spectrometer (Varian, Palo Alto, CA, USA) equipped with a GladiATR diamond attenuated total reflectance (ATR) accessory. This spectrometer features a high-sensitivity dual-detector configuration with mercury cadmium telluride (MCT) and deuterated triglycine sulfate (DTGS) detectors. Measurements were carried out in the wavenumber range of 400–4000 cm^−1^. Each infrared spectrum was obtained by averaging 128 scans at a resolution of 4 cm^−1^. Data acquisition and analysis were performed using the Varian Resolutions Pro software version 5.1.0.829. Although baseline correction was not applied, all spectra were recorded under identical conditions, enabling a reliable qualitative comparison of band intensities.

A JEOL JSM-7600F high-resolution scanning electron microscope (Tokyo, Japan) with a secondary electron imaging (SEI) detector was used for surface characterization. The topography of all samples was examined at an accelerating voltage of 10.0 kV and a working distance of approximately 6.0 mm.

The specific surface area of the pyrochars was determined using nitrogen adsorption–desorption isotherms at 77.3 K, measured with a Quantachrome NOVA Station A surface area and porosity analyzer. Before the measurements, the samples were degassed under vacuum at 200 °C for 72 h to remove moisture and adsorbed gases. This degassing temperature may be insufficient to fully remove strongly bound volatiles from high-temperature pyrochars, potentially leading to a slight underestimation of the surface area, as further discussed in the Results Section.

The BET surface area was calculated using the multipoint Brunauer–Emmett–Teller (BET) method over a relative pressure (P/P_0_) range of 0.05–0.10. The data were processed with the NovaWin software (version 11.06). Results are reported as mean values from three independent replicates (*n* = 3).

### 2.5. Batch Adsorption Experiments

Kinetic and preliminary adsorption tests were carried out by adding 5.0 mL of the metribuzin solution (23.1 mg/L) to individual 10 mL glass vials containing 0.010, 0.015, 0.020, 0.025, 0.050, or 0.075 g of biochar or hydrochar samples. The vials were sealed and shaken at laboratory temperature (25 ± 1 °C) at 50 rpm using a rotary shaker (tube roller RSLAB-10, Auxilab, Navarra, Spain), and shaking times ranged from 0 to 12 h. The suspensions were centrifuged (1200 rpm, 5 min) to separate the solid and liquid phases. The supernatant was collected, filtered through a 0.45 μm PTFE filter, and analyzed for residual metribuzin concentration using high-performance liquid chromatography (Agilent 1260 Infinity II HPLC with MWD Detector).

The control samples, which contained metribuzin without any sorbent, maintained a constant concentration throughout the experiment. This outcome confirms that the sorption observed in the test samples was entirely due to interactions with the biochar surface.

Equilibrium adsorption isotherms were investigated under the same conditions at laboratory temperature (25 ± 1 °C), using a 5.0 mL metribuzin solution with initial concentrations ranging from 5.0 to 35.0 mg/L and 50.0 mg of the adsorbent. Based on the equilibrium point determined from kinetic experiments, the contact time was fixed at 9 h. The same centrifugation and analytical procedures were applied as described for the kinetic experiments.

All experimental conditions were tested in triplicate (*n* = 3) to ensure data consistency and reproducibility. Due to the low standard deviations between replicates, error bars are not included in the graphical presentation of the results. Standard deviations for all measurements were below ±3%, confirming high reproducibility.

### 2.6. Sorption Data Analyses

The equilibrium adsorption capacity of metribuzin on the hydrochars or biochars *Q*_e_ (mg/g) was calculated using the mass balance equation [45]:*Q*_e_ = ((*C*_e_ − *C*_0_) × *V*)/*m*(1)
where *C*_0_ is the initial concentration of the sorbate (mg/L), *C*_e_ is the equilibrium concentration of the sorbate after treatment (mg/L), *V* is the volume of the metribuzin solution (L), and *m* is the mass of the adsorbent (g).

Adsorption kinetics were evaluated using pseudo-first-order and pseudo-second-order models. The pseudo-first-order model, proposed by Lagergren and commonly applied in sorption studies [46], is given by the following:(*dQ*_t_/*dt*) = *k*_1_ × (*Q*_e_ − *Q*_t_)(2)

Or, it can be expressed as follows in linear form:log(*Q*_e_ − *Q*_t_) = log*Q*_e_ − *k*_1_ × *t*(3)
where *Q*_e_ is the adsorption capacity at equilibrium (mg/g), *Q*_t_ is the amount of metribuzin adsorbed at time *t* (mg/g), and *k*_1_ is the pseudo-first-order rate constant (1/min). This model assumes that the adsorption rate is proportional to the number of unoccupied active sites [47].

The pseudo-second-order kinetic model, developed by Ho and McKay [48], assumes chemisorption as the rate-limiting step and is expressed as follows:(*dQ*_t_/*dt*) = *k*_2_ × (*Q*_e_ − *Q*_t_)^2^(4)

Or, it can be expressed as follows in linear form:*t*/*Q*_e_ = [1/(*k*_2_ × *Q*_e_^2^)] + (*t*/*Q*_e_)(5)
where *k*_2_ is the pseudo-second-order rate constant (g/mg·min). This model describes systems where chemisorption controls adsorption, involving electron sharing or exchange between adsorbate molecules and surface functional groups [49].

Equilibrium data were analyzed using the Langmuir and the Freundlich isotherm models. The Langmuir model assumes monolayer adsorption on a homogeneous surface with identical adsorption sites and no interaction between adsorbed molecules. It is expressed as follows [50]:*Q*_e_ = (*Q*_max_ × *K*_L_ × *C*_e_)/(1 + *K*_L_ × *C*_e_)(6)

Or, it can be expressed as follows in linear form:*C*_e_/*Q*_e_ = *C*_e_/*Q*_max_ + 1/(*K*_L_ × *Q*_max_)(7)
where *Q*_max_ (mg/g) is the maximum adsorption capacity, and *K*_L_ (L/mg) is the Langmuir constant. The dimensionless separation factor, *R*_L_, is defined as follows [51]:*R*_L_ = 1/(1 + *K*_L_ × *C*_0_)(8)
where *K*_L_ is the Langmuir constant (L/mg), and *C*_0_ is the highest initial concentration of the adsorbate (mg/L).

This was used to assess the favorability of adsorption, where *C*_0_ is the highest pesticide concentration (mg/L), with *R*_L_ > 1 indicating unfavorable adsorption, *R*_L_ = 1 linear, 0 < *R*_L_ < 1 favorable, and *R*_L_ = 0 irreversible.

The Freundlich isotherm is an empirical model suitable for heterogeneous surfaces and non-uniform adsorption energies. It is expressed as follows:*Q*_e_ = *K*_F_ × *C*_e_^1/n^(9)

Or, it can be expressed as follows in linear form:log*Q*_e_ = log*K*_F_ + 1/n × log*C*_e_(10)

In the Freundlich model, *K*_F_ is the adsorption capacity constant, and 1/n is an empirical parameter that describes the adsorption intensity. Values of 1/n < 1 indicate favorable adsorption conditions, while values of 1/n > 1 suggest adsorption onto a highly heterogeneous surface or reduced binding strength [50].

All calculations and model fittings were performed using Microsoft Excel 2016.

### 2.7. HPLC-UV Analysis of Metribuzin

The concentration of metribuzin was determined using a previously developed and validated high-performance liquid chromatography method with UV detection (Agilent 1260 Infinity II HPLC with MWD Detector), as described in [52]. An Agilent InfinityLab Poroshell 120 Eclipse EC-C18 column with a pore size of 4 μm, an inner diameter of 4.6 mm, and a length of 150 mm was used (Agilent Technologies, Santa Clara, CA, USA). Chromatographic separation was performed under isocratic conditions using a mobile phase consisting of ultrapure water and acetonitrile in a 50:50 (*v*/*v*) ratio (ultrapure water:acetonitrile). The flow rate was set to 1.0 mL/min, with an injection volume of 10 μL. The separation was carried out at a column temperature of 50 ± 0.5 °C. The detection wavelength was set to 254 nm, corresponding to the UV absorption maximum of metribuzin, which was determined using an Agilent Cary 60 UV-Vis spectrophotometer.

## 3. Results and Discussion

### 3.1. Ash Content in Chlorella vulgaris Biomass

Ash content is a crucial parameter in evaluating the suitability of biomass for energy recovery and technological processing. This study determined the ash content of dried *Chlorella vulgaris* biomass [43]. The average value from six measurements was 25.01 ± 0.77% (*w*/*w*). The higher proportion of inorganic substances in algae is related to their ability to accumulate minerals from the culture medium.

Published values vary considerably depending on the type of microalgae, the culture medium’s composition, and the sample processing method. Bach et al. [53] reported 33.2% for *Chlorella vulgaris* ESP-31, while Grierson et al. [54] reported 15.1–27.0% for different species and growth stages. Jabeen et al. [55] measured only 5.3% after hydrogen peroxide treatment, reducing the residual carbon content. Bumbac et al. [56] report 4.54 ± 0.74% for commercially processed biomass, where thorough rinsing and removal of mineral salts is assumed.

The notably higher value achieved in the present study is probably related to the use of biomass grown without subsequent cleaning, as well as to the standard combustion procedure according to the standard. The different values indicate that the selection of biomass concerning ash content can influence the final properties of biochar. Although the ash content was not determined in the final chars, the high ash level measured in the raw *Chlorella vulgaris* biomass (25.01%) likely influenced the physicochemical properties of the resulting hydrochars and pyrochars. In hydrochars, mineral components likely contributed to the presence of polar functional groups, as indicated by FTIR spectra, and enhanced metribuzin adsorption through either electrostatic interactions or surface complexation. Despite the potentially pore-blocking nature of ash, hydrochars prepared at 200 °C—derived from the same high-ash biomass—achieved the highest adsorption capacity. This suggests that in this case, the chemical surface functionality played a more decisive role compared to porosity.

### 3.2. Yields of Hydrochars and Pyrochars from Microalgae

A biochar yield represents the efficiency of the thermochemical conversion of biomass. The yields of hydrochars and pyrochars produced from *Chlorella vulgaris* varied significantly depending on the processing temperature, as shown in Figure 1.

During hydrothermal carbonization (HTC) in a PPL100 hydrothermal autoclave reactor using 6.00 g of biomass and 60 mL of distilled water, the highest solid yield was achieved at 200 °C (33.32%). At the same time, at 250 °C, it decreased to 25.76%. This process was carried out at a heating rate of 5 ± 0.5 °C/min with a residence time of 60 min. After cooling, the solid residue was separated by centrifugation, washed, dried at 105 °C, and stored in a desiccator. The decreasing trend in yield with an increasing temperature was caused by increased solubility and degradation of biomass components under subcritical water conditions [57,58].

Pyrochar yields were determined after slow pyrolysis of 5.00 g of dried biomass at temperatures ranging from 300 °C to 500 °C, with a heating rate of 10 °C/min, under continuous nitrogen flow (20 mL/min) and a residence time of 60 min. The highest pyrochar yield was recorded at 300 °C (63.56%) and gradually decreased with an increasing temperature to 27.21% at 500 °C. This trend aligns with the known thermal decomposition of biomass, where higher temperatures lead to more intense devolatilization and thus to a lower yield of solid residue [57,59,60,61].

Other authors also reported comparable values. For example, Grierson et al. [54] reported a pyrochar yield of 34% at 500 °C for *Chlorella vulgaris*. Binda et al. [28] reported a pyrochar yield ranging from 55% to 29% as the temperature increased from 300 °C to 500 °C. As for HTC, Park et al. [27] achieved a hydrochar yield of 32% at 200 °C using *Chlorella vulgaris*, which is consistent with the value obtained in this study at 200 °C.

The relatively high pyrochar yield observed at 300 °C indicates that low-temperature pyrolysis may be suitable for studies aiming to maximize carbon yields in the solid phase.

### 3.3. Active and Exchangeable pH of Biomass and Carbonized Products

The pH of biochars significantly impacts their suitability for environmental applications, particularly in terms of pollutant adsorption and soil improvement. In this study, two types of pH were measured: active pH, which reflects the concentration of freely dissociable hydrogen ions in an aqueous environment, and exchangeable pH, which includes hydrogen ions displaced from the sorbent surface by a neutral salt (KCl), thereby better representing the potential acidity relevant to sorption interactions.

This study measured the active and exchangeable pH of dried *Chlorella vulgaris* biomass, hydrochars produced at 200 and 250 °C, and pyrochars prepared at temperatures ranging from 300 to 500 °C. The results are shown in Figure 2 as a function of treatment temperature.

The raw biomass showed an active pH of 6.09 and an exchangeable pH of 5.91, indicating near-neutral surface properties with a slightly lower ion exchange capacity. Hydrochars exhibited lower values: the active pH ranged from 4.49 to 5.15 and the exchangeable pH from 4.67 to 5.19. Such low pH values are characteristic of HTC materials, resulting from the formation of oxygen-containing functional groups and the relatively low ash content, which limits their buffering capacity.

As the process temperature increased during pyrolysis, active and exchangeable pH values also increased. Pyrochars exhibited a progressive shift toward more neutral or slightly alkaline conditions, with active pH values ranging from 6.02 (at 300 °C) to 6.88 (at 500 °C) and exchangeable pH values from 5.22 to 4.87. Although the exchangeable pH remained lower than the active pH across all samples, the difference between the two increased with temperature. This likely reflects the progressive thermal degradation of acidic surface groups alongside an increasing contribution of alkaline-forming mineral components.

Other authors have reported similar trends. Park et al. [27] observed a pH of 3.85–4.84 for hydrochars and 5.84–6.40 for pyrochars derived from *Chlorella vulgaris*, confirming the increasing pH trend with a rising temperature. In contrast, Tag et al. [62] documented even higher pH values, up to 13.7, for algal biochars produced at temperatures ranging from 250 to 600 °C, attributing these results to the accumulation of alkaline minerals and significant increases in ash content. Such differences underline the influence of feedstock origin, pretreatment, and carbonization conditions on the final acid–base properties.

The increasing trend in pH observed in this study supports the general understanding that pyrolysis leads to the loss of carboxylic and phenolic groups and the enrichment of basic mineral components [55]. Such pH characteristics must be considered when evaluating the applicability of algal biochars in environmental applications, particularly for soil pH adjustments or the adsorption of ionic pollutants.

This trend also aligns with FTIR results, which show a progressive loss of acidic oxygen-containing functional groups (–COOH and –OH) at higher pyrolysis temperatures. Although the ash content was only measured in raw biomass (25.01%), it likely contributed to the buffering and gradual pH increase in pyrochars, as basic mineral components (e.g., carbonates, alkali, and alkaline earth metals) concentrate during thermal treatments.

### 3.4. Oxidizable Organic Carbon Content

Thermochemical processing affects a material’s amount and stability, typically increasing the proportion of stable carbon forms at higher temperatures. The C_org_ content of all samples was measured to compare the preservation of oxidizable carbon. The oxidizable organic carbon (C_org_) content in the raw biomass and carbonized samples was determined using the Turin method, modified according to Nikitin. The C_org_ values are summarized in Table 1.

The untreated *Chlorella vulgaris* biomass contained 375.13 mg/g of oxidizable carbon. A slight increase was observed in hydrochars produced at 200 °C (384.74 mg/g), with a more pronounced rise at 250 °C (487.95 mg/g), likely due to volatile loss and the partial enrichment of stable carbon forms during hydrothermal processing. Among the pyrochars, the highest C_org_ content was observed at 400 °C (625.13 mg/g), followed by 500 °C (528.33 mg/g) and 350 °C (477.69 mg/g). These results suggest that moderate pyrolysis temperatures (350–400 °C) favor preserving carbon in the solid phase of algal biomass.

Comparable C_org_ levels have been reported in earlier studies. Law et al. [63] documented 62.0% carbon in *Chlorella vulgaris* pyrochars formed at 350 °C, while Yu et al. [64] reported 57.4% carbon for wet-torrefied material formed at 170 °C. Costa et al. [65] measured 61.3% carbon in pyrochars obtained from commercially sourced *Chlorella vulgaris*. Hydrochar products generally retain lower carbon fractions—for instance, Yu et al. [64] reported 27.0% carbon following hydrothermal liquefaction at 350 °C. Similarly, pyrolysis of low-lipid *Chlorella vulgaris* at 500 °C yielded 53.5% carbon, and *Spirulina platensis* biochars produced at the same temperature exhibited a C_org_ content of approximately 51.0% [66].

### 3.5. FTIR Analysis of Functional Group Changes During Thermal Treatment of Chlorella vulgaris

FTIR spectra of raw *Chlorella vulgaris* biomass, hydrochars produced at 200 °C and 250 °C, and pyrochars formed at 300–500 °C show distinct changes in functional groups depending on the treatment temperature (Figure 3). Spectra of hydrochars appeared slightly noisier compared to those of pyrochars, likely due to residual moisture and structural heterogeneity, which can reduce peak sharpness and signal intensity. While aromatic structures are often associated with well-defined FTIR bands in carbonaceous materials, the peak shape and intensity are also strongly influenced by factors such as molecular order, degree of conjugation, and sample homogeneity. In the case of hydrochars, the presence of partially decomposed biomolecules, polar functional groups, and amorphous carbon may contribute to broader and less intense peaks, rather than lower aromaticity alone. The results were compared with the existing literature and reflect processes such as decarboxylation, dehydration, and aromatization, occurring during hydrothermal carbonization (HTC) and pyrolysis.

In the raw biomass, a broad band between 3600 and 3200 cm^−1^ corresponds to O–H and N–H stretching vibrations from hydroxyl, carboxylic, and amide groups. This band decreases in hydrochars and nearly disappears in pyrochars above 300 °C. This reduction is associated with the release of water and the degradation of hydroxyl- and amide-containing compounds [27,28]. In contrast, Khoo et al. [67] reported an increase in this region in hydrochars, which may be due to differences in the moisture content or algal composition.

Stretching vibrations of aliphatic CH_3_ and CH_2_ groups appear at 2970–2850 cm^−1^. These bands are weak in the raw biomass but increase in hydrochars (particularly at 200 °C), indicating partial preservation of aliphatic structures. At temperatures of ≥350 °C, the bands diminish due to the thermal degradation of aliphatic chains [27,28,53,67,68].

A visible band in the 2550–2540 cm^−1^ region was observed in some FTIR spectra (Figure 3). According to Liu et al. [61], this band may be related to the presence of thermally sensitive carboxylic groups. Furthermore, the possibility of either instrumental or atmospheric influences (e.g., CO_2_) cannot be ruled out, as similar bands often occur due to measurement conditions and are not typical of the main functional groups in biochars [69]. Therefore, both possibilities were considered in the interpretation of the results. Weak absorption bands occasionally observed between 2400 and 2300 cm^−1^ likely arise from atmospheric CO_2_, which can interfere during ATR-FTIR measurements. These features are not related to the sample composition and were excluded from the interpretation of the results.

In the carbonyl region (1745–1700 cm^−1^), attributed to C=O stretching in esters and carboxylic acids, signals are distinct in the biomass and hydrochars but fade or disappear in pyrochars formed above 300 °C due to a decomposition of these functional groups [67,70].

The 1655–1540 cm^−1^ region shows a transition from amide I (~1655 cm^−1^) to aromatic C=C (~1600 cm^−1^), indicating the conversion of protein-derived amide groups to conjugated aromatic structures. These signals remain visible in pyrochars, suggesting nitrogen-containing aromatic products [27,53,68].

In the 1510–1450 cm^−1^ range, an increasing band intensity correlates with the degradation of aliphatic structures and the formation of aromatic rings at higher temperatures, especially above 400 °C. This trend aligns with the formation of condensed aromatic structures [27].

The weak band at 1245–1230 cm^−1^, attributed to asymmetric P=O stretching, appears in the raw biomass but disappears after carbonization. This likely results from phosphate degradation or solubilization; however, this interpretation is uncertain due to overlapping signals [70].

In the 1200–1000 cm^−1^ range, a strong signal attributed to polysaccharides and cellulose-like structures dominates the biomass spectrum; however, it decreases sharply after treatment at 200 °C. It becomes nearly undetectable in pyrochars, indicating depolymerization of carbohydrates [27,28,53,67].

Below 900 cm^−1^, bands corresponding to aromatic C–H out-of-plane bending become more pronounced with an increasing temperature, supporting the formation of condensed aromatic structures. Signals between 620–580 cm^−1^ may arise from residual minerals or additional aromatic deformation bands [27].

The FTIR spectra indicate a transformation of hydroxyl (3600–3200 cm^−1^), carboxyl (1745–1700 cm^−1^), and amide groups (around 1655 cm^−1^) as a function of treatment temperature; however, the loss is not strictly gradual. The hydroxyl band is clearly present only in the raw biomass, while the carboxyl band shows comparable intensity in pyrochars formed at 450 and 500 °C. Moreover, due to a partial spectral overlap, these changes are not always clearly distinguishable without normalization or peak deconvolution. These signals are still present in hydrochars produced at 200–250 °C but nearly vanish in pyrochars above 300 °C. Meanwhile, bands assigned to aromatic C=C stretching (1600–1450 cm^−1^) and out-of-plane C–H bending (<900 cm^−1^) become stronger, reflecting the development of nitrogen-containing aromatic and condensed ring structures. As a result, pyrochars exhibit a more aromatic and less polar surface. In raw biomass and hydrochars, the sorption is likely driven by hydrogen bonding and electrostatic interactions, whereas in pyrochars, π–π and hydrophobic interactions become more significant. These structural changes alter the surface chemistry of the chars in ways that affect sorption mechanisms. In the case of metribuzin—a moderately polar compound with both aromatic and heteroatom-containing functional groups—hydrogen bonding may dominate in hydrochars, while π–π stacking and hydrophobic interactions likely play a more prominent role in pyrochars.

Although a CHN analysis was not performed, the FTIR spectra and known composition of *Chlorella vulgaris* suggest that the materials primarily contain C; H; O; and nitrogen-bearing functionalities, such as amides [71].

### 3.6. Microstructure of Raw Biomass, Hydrochars, and Pyrochars

The microstructure and porosity of raw biomass, hydrochars, and pyrochars were examined using scanning electron microscopy (SEM). Before imaging, the samples were sputter-coated with a conductive layer to prevent excessive surface charging when exposed to the electron beam. Representative images of raw dried biomass and hydrochars are shown in Figure 4. The porosity of materials could not be evaluated precisely by image analysis due to charging artifacts. Therefore, only a qualitative description of the morphology is provided in the following paragraphs.

In the untreated biomass, the algal cells formed a compact and interconnected structure with limited visible porosity. In contrast, hydrochars exhibited greater porosity. The hydrochar prepared at 200 °C exhibited partial retention of the original biomass morphology, characterized by a roughened texture and scattered surface cavities, likely formed during the HTC process through degassing. While some cellular contours were still observable, the structure was less dense than in raw biomass.

The hydrochar produced at 250 °C consisted of disintegrated cellular fragments with no continuous matrix, suggesting thermal disruption and collapse of the microalgal structure at this temperature. The structure appeared fragmented, composed primarily of disintegrated cell residues. This structural breakdown was attributed to more intense devolatilization and higher thermal stress during the process.

According to Park et al. [27], hydrochars produced at 170 °C formed aggregated microspheres with a rough surface, while materials treated at 200 °C exhibited irregular clusters with wrinkled textures. In comparison, pyrochars obtained at 300–400 °C displayed fragmented surfaces with sheet-like structures and well-developed porosity. These morphological changes reflect a progressive decomposition of the original biomass and the increasing degree of carbonization at elevated temperatures.

A microscopic analysis at 5000× magnification (Figure 5) revealed a progressive disruption of cellular structures with an increasing temperature. The hydrochar produced at 200 °C displayed a heterogeneous surface with residual structural features and fragmented textures. At 250 °C, the material appeared more compact but showed extensive disintegration into irregular particles. In contrast, pyrochars produced at 300 °C and 350 °C retained more defined surface features, with smooth and consolidated structures, suggesting a slower decomposition of cellular matter. At higher temperatures (400 °C and 500 °C), the material exhibited advanced fragmentation and the loss of morphological integrity, indicating intensified carbonization.

The observed differences arise from the processing environment. In hydrothermal carbonization, subcritical water acts as both a solvent and a reactant, breaking down cellular structures and causing earlier disintegration of the microalgal biomass. In comparison, slow pyrolysis proceeds in a dry, inert environment, which delays structural collapse and promotes the gradual enrichment of carbon. Previous studies have reported similar contrasts, describing hydrochars as more fractured and porous while noting that pyrochars retained a more compact morphology [28].

Pyrolysis exposes organic matter to high temperatures in the absence of oxygen, making it an anaerobic thermochemical process. In this environment, the microalgal biomass is heated to the target temperature, resulting in cell swelling due to internal gas formation. However, the absence of oxygen slows the degradation reactions. As the temperature increases, thermal decomposition becomes more pronounced due to accelerated reaction kinetics.

As shown in Figure 5, significant structural changes in pyrochars begin to appear at temperatures ≥ 400 °C. At this point, the biomass undergoes both physical breakdown and a chemical transformation, resulting in the formation of carbon-rich material with altered morphology. These findings are consistent with observations reported by Sotoudehniakarani et al. [72], who noted that *Chlorella vulgaris* retained much of its cellular structure during pyrolysis below 400 °C, whereas substantial collapse and disordered carbon phase formation occurred only at higher temperatures (≥450 °C). Similarly, Park et al. [27] reported that notable porosity development and structural disruption in algal-derived pyrochars became evident only above this temperature threshold.

### 3.7. Specific Surface Area and BET Parameters of Pyrochars

The specific surface areas of selected pyrochar samples were determined by nitrogen adsorption using the classical Brunauer–Emmett–Teller (BET) [73] method at 77 K. The results for samples produced at different pyrolysis temperatures are summarized in Table 2.

Samples prepared at 300 to 450 °C exhibited very low BET surface areas, whereas the surface area increased markedly at 500 °C, reaching 44.3 m^2^/g. The sample PCH 300 exhibited a surface area of 0.332 m^2^/g, along with a relatively high BET C constant (228.7), indicating a limited number of active sites capable of stronger interactions. The surface area for PCH 350 increased to 2.10 m^2^/g, though its low C constant (6.77) may reflect weaker interactions between the adsorbate and surface functionalities. PCH 400 showed a similarly low surface area (0.688 m^2^/g) with a moderately high C value (121.8), while PCH 450 reached 3.20 m^2^/g, with a C constant of 7.30. A significant increase in surface area was observed only for PCH 500, which displayed 44.3 m^2^/g, likely resulting from enhanced devolatilization and pore development at elevated temperatures.

Despite the relatively similar external morphology of the PCH 300–450 samples observed in the SEM images (Figure 5), the BET surface areas and C constants vary within 1–2 orders of magnitude. These discrepancies can be attributed to differences in surface chemistry and pore accessibility rather than macroscopic morphology. The BET C constant is related to the strength of adsorbate–adsorbent interactions, where higher values typically indicate either more polar or energetically heterogeneous surfaces [73]. For example, PCH 300 exhibited a low surface area but a high C constant, suggesting the presence of strong binding sites with limited accessibility. In contrast, PCH 450 had a larger surface area but a lower C constant, indicating weaker interactions over a more open surface. This inverse relationship between C and surface area may reflect the progressive loss of polar surface groups during pyrolysis, which reduces the interaction strength while promoting pore opening and surface development.

All samples exhibited BET linear fit correlation coefficients above 0.998, indicating a good fit of the data to the BET model and a reliable calculation of surface parameters.

Compared to similar studies, the surface areas obtained here were lower despite comparable processing temperatures. For instance, Park et al. [27] reported BET surface areas up to 17.0 m^2^/g for a *Chlorella vulgaris* pyrochar prepared at 600 °C. In another study, Nejati et al. [74] reported a BET surface area of 15.45 m^2^/g for a *Chlorella vulgaris*-derived biochar obtained at 650 °C, which increased up to 505.10 m^2^/g after KOH activation. Law et al. [63] also noted that microalgal biochars obtained at temperatures above 500 °C can, in some cases, reach surface areas comparable to those of activated carbon, especially when the mineral content is low and the material becomes highly carbonized. Similarly, Gong et al. [75] observed that the BET surface area of a *Chlorella vulgaris* biochar increased with a rising pyrolysis temperature. However, the values remained relatively low, with a maximum below 1.5 m^2^/g, even at 700 °C. These variations likely reflect differences in biomass compositions; process parameters; and methodological factors, including degassing conditions, analysis temperatures, data processing, the choice of BET fitting ranges, and the applied isotherm model, all of which can influence the reported surface areas and C constants.

The BET method using nitrogen may underestimate surface areas in microalgal chars due to limited access to micropores, especially in materials rich in polar surface groups. Nitrogen is commonly used as a standard adsorbent. However, its interaction with biochars rich in polar functional groups, such as those derived from proteinaceous or carbohydrate-rich biomass, may be limited. These surface properties can restrict N_2_ access to narrow micropores or underestimate the accessible surface area. This issue has been discussed in previous studies, where the low BET surface area of microalgal biochars was attributed to their chemical and morphological characteristics. Furthermore, it has been reported that pyrolysis at higher temperatures reduces the content of oxygenated surface groups, potentially enhancing N_2_ accessibility and improving apparent surface area readings. For this reason, BET results for pyrochars, particularly those derived from microalgal biomass, should be interpreted cautiously. Alternative techniques, such as CO_2_ adsorption, may provide a more accurate assessment of microporosity in these materials.

### 3.8. Effect of Biomass Dose

The effect of biochar dosage on metribuzin removal is presented in Figure 6. Sorption efficiency improved as the adsorbent dose increased from 10 to 50 mg, likely due to the greater number of available surface sites participating in metribuzin binding [76].

Hydrochars removed more metribuzin compared to pyrochars at all tested dosages. This was likely due to the higher abundance of surface functional groups described in the previous section. Among the tested samples, the hydrochar produced at 200 °C achieved the highest removal efficiency, exceeding 30% at a dose of 50 mg. At 75 mg, the removal efficiency decreased slightly. It is assumed that particle agglomeration occurred, leading to a reduction in the available surface area, thereby slowing down the mass transfer process itself [77]. Therefore, we used 50 mg of the adsorbent in all subsequent adsorption tests. The reduction in the available surface area is planned to be verified in a separate study by laser diffraction methods.

### 3.9. Effect of Contact Time

We examined how contact time influences the sorption efficiency of hydrochars (HCHs) and pyrochars (PCHs) over a period ranging from 15 to 720 min at 25 °C. As shown in Figure 7, adsorption equilibrium was generally reached between 6 and 12 h. Based on these results, a contact time of 9 h was selected for subsequent sorption experiments.

We evaluated the experimental data using pseudo-first-order and pseudo-second-order kinetic models. Table 3 summarizes the model parameters, including the rate constants (*k*_1_ and *k*_2_), the experimental and calculated adsorption capacities (*Q*_e_ exp and *Q*_e_ calc), and the correlation coefficients (R^2^).

The pseudo-second-order model provided a better fit for most samples, as reflected by the higher R^2^ values. This result suggests that chemisorption is likely the dominant mechanism involved in metribuzin adsorption, involving either hydrogen bonding, as metribuzin contains amino and carbonyl groups that can form hydrogen bonds with surface functional groups, such as –OH and –COOH, or electron sharing/exchange, e.g., covalent or coordination bonding through lone pairs on nitrogen or oxygen. The surface complexation involving other functional groups present on the adsorbent may also contribute to the adsorption process. The pseudo-first-order model showed a slightly better fit only in the case of the hydrochar produced at 250 °C, indicating that physisorption might play a more significant role under these specific conditions. This model typically describes the initial stages of adsorption more accurately but often fails to capture the behavior in later stages, where the rate is limited by either diffusion or weaker interactions.

### 3.10. Adsorption Isotherms

We used the Langmuir and Freundlich isotherm models to evaluate the relationship between the amount of metribuzin adsorbed and its equilibrium concentration in solution. Table 4 summarizes the calculated isotherm parameters, and Figure 8 shows the fitted adsorption curves.

The correlation coefficients (R^2^) indicated a better fit with the Freundlich model for all tested sorbents, suggesting adsorption on heterogeneous surfaces with varying binding energies. The surface heterogeneity and presence of functional groups are discussed in more detail in the chapters on the SEM and FTIR results. The analysis of the Freundlich constant 1/n revealed favorable adsorption (1/n < 1) for HCH 200 °C, PCH 400 °C, PCH 450 °C, and PCH 500 °C, indicating that adsorption increases with concentration, though at a decreasing rate. In contrast, 1/n > 1 for HCH 250 °C, PCH 300 °C, and PCH 350 °C suggests an unfavorable adsorption behavior, where sorption efficiency decreases with an increasing concentration.

In contrast, the Langmuir model provided a poorer fit. This result suggests that adsorption likely does not occur on a homogeneous surface with identical active sites, and monolayer coverage is not achieved. For PCH 300 °C and PCH 350 °C, the Langmuir model even yielded non-physical (negative) values for *Q*_max_ and *K*_L_; therefore, the corresponding isotherms were omitted from Figure 8. This phenomenon may occur when an adsorption site can bind more than one adsorbate molecule and interactions exist between adsorbed molecules. The Langmuir separation factor, *R*_L_, was also calculated and is presented in Table 4. All *R*_L_ values fell within the range of 0 < *R*_L_ < 1, indicating favorable adsorption. However, their very low values suggest stronger sorbent–adsorbate interactions and a tendency toward irreversibility, especially at higher concentrations. A detailed desorption study would be necessary to confirm this assumption.

At an equilibrium concentration of 20 mg/L, HCH 200 °C, PCH 400 °C, and PCH 500 °C exhibited the highest adsorption capacities, making them the most effective sorbents in this study.

## 4. Conclusions

This study evaluated *Chlorella vulgaris* biochars prepared via hydrothermal carbonization (200 °C and 250 °C) and slow pyrolysis (300–500 °C) for the removal of metribuzin from aqueous solutions. The characterization included the yield, ash content, pH, total organic carbon (TOC), Brunauer–Emmett–Teller (BET) surface area, Fourier Transform Infrared (FTIR) spectra, and scanning electron microscopy (SEM) imaging. Raw biomass served as a baseline for comparison. The hydrochar produced at 200 °C exhibited the highest adsorption capacity of all tested sorbents. This performance was linked to the retention of cell morphology (confirmed by SEM; Figure 4b), a high density of oxygenated functional groups (confirmed by FTIR; Figure 3), and favorable accessibility, despite the lack of porosity data. In contrast, increasing the HTC temperature to 250 °C resulted in degradation of the structure and surface functionality, thereby reducing adsorption efficiency. The most effective pyrochar was produced at 500 °C. Despite its high surface area (44.3 m^2^/g) and well-developed porosity, it failed to surpass the adsorption capacity of the 200 °C hydrochar, likely due to the loss of polar groups essential for pesticide binding. Pyrochars prepared at 300–450 °C retained partial structural integrity but showed limited surface development and weaker adsorption performance. Kinetic modeling confirmed that metribuzin adsorption followed pseudo-second-order behavior, suggesting that chemisorption is the dominant mechanism. The equilibrium data align more closely with the Freundlich isotherm, indicating multilayer adsorption on heterogeneous surfaces. The hydrochar prepared at 200 °C proved to be the most effective sorbent under the tested conditions. Its low production temperature, functional group richness, and intact cell structure contributed to this outcome. Although porosity was not quantified for hydrochars, their adsorption performance suggests that surface chemistry played a dominant role. *Chlorella vulgaris*, although not utilized in this manner in the present study, can be cultivated in wastewater environments, thereby contributing to nutrient recovery and CO_2_ capture. This further highlights its relevance in circular bioeconomy strategies. While some studies [21,26,78] reported higher adsorption capacities for metribuzin using biochars or hydrochars produced under optimized or modified conditions, these materials were typically derived from lignocellulosic biomass, such as wood, straw, or manure, rather than from microalgae, like *Chlorella vulgaris*. Additionally, many of these sorbents required higher processing temperatures or chemical activation steps, which increase production costs and energy demands. In contrast, the biochars examined here were produced without surface modifications, using renewable feedstock and employing simple thermal methods. These properties support their potential use in decentralized and affordable water treatment technologies. Researchers have investigated *Chlorella*-derived sorbents for the removal of dyes, heavy metals, and pharmaceuticals [23,79,80,81]. Studies focusing on pesticide removal, especially metribuzin, remain rare. This contribution broadens the understanding of algae-based biochars and supports their further development as functional materials for environmental remediation. While the adsorption capacity was lower than that of some chemically activated sorbents reported in the literature, the unmodified biochars produced here offer practical advantages in terms of simplicity, cost, and environmental sustainability. This makes them suitable candidates for decentralized or low-tech water treatment systems, where low energy input and renewable feedstocks are prioritized. This study presents a novel application of hydrochars and pyrochars derived from a single microalgal feedstock under two distinct thermal conversion pathways for metribuzin removal. The integration of kinetic experiments and isotherm modeling further strengthens the mechanistic interpretation of adsorption. Future research should focus on modifying these sorbents through activation or composite formation to improve performance. In addition, future studies should investigate desorption behavior and advanced surface characterization (e.g., CO_2_ adsorption for microporosity) and should be combined with computational modeling of adsorbate–adsorbent interactions to deepen our understanding of the mechanisms involved. Nonetheless, even in their unmodified state, microalgae-derived biochars demonstrate potential for their integration in sustainable water treatment systems.

## Figures and Tables

**Figure 1 materials-18-03374-f001:**
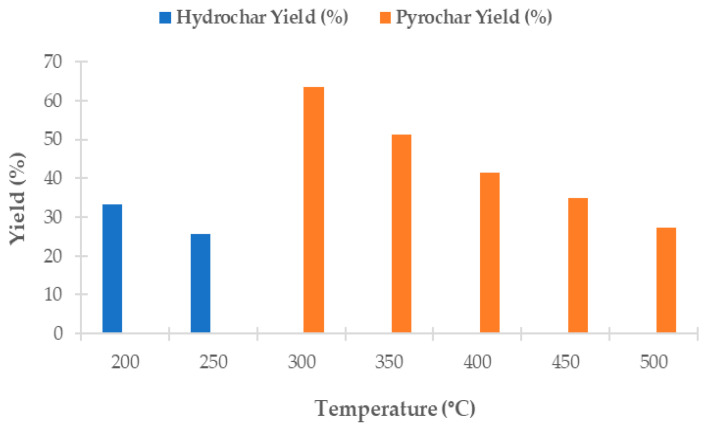
Yields of hydrochars and pyrochars obtained from *Chlorella vulgaris* at different processing temperatures.

**Figure 2 materials-18-03374-f002:**
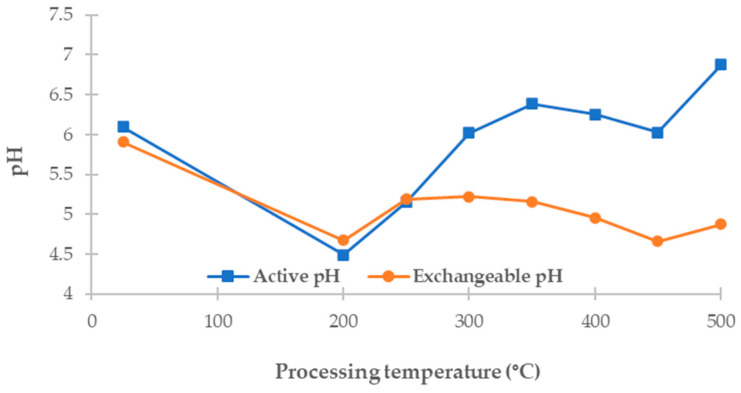
Active and exchangeable pH values of untreated biomass (plotted at 25 °C), hydrochars (200–250 °C), and pyrochars (300–500 °C) produced from *Chlorella vulgaris*, concerning processing temperature.

**Figure 3 materials-18-03374-f003:**
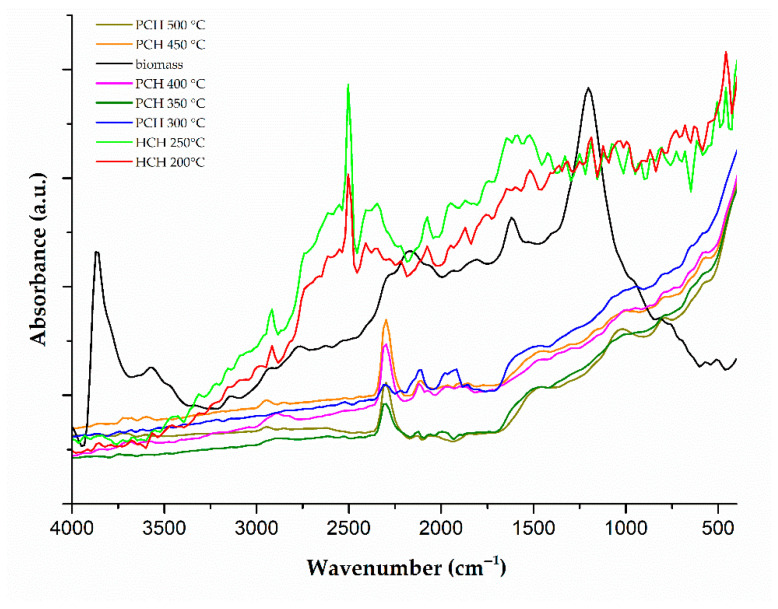
FTIR spectra of *Chlorella vulgaris* biomass, hydrochars (HCHs) produced at 200 °C and 250 °C, and pyrochars (PCHs) obtained at 300–500 °C.

**Figure 4 materials-18-03374-f004:**
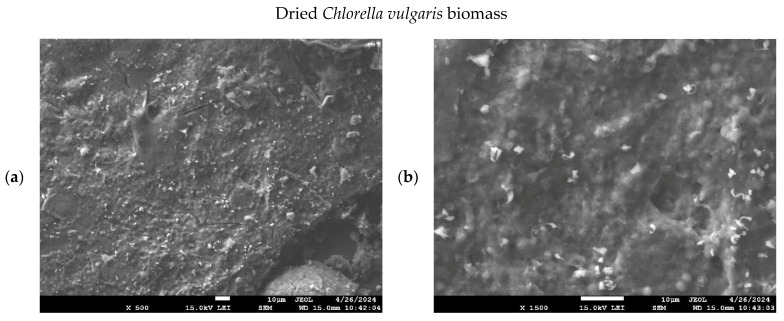
Microstructure of microalgal cells in raw biomass at 500× magnification (**a**), raw biomass at 1500× magnification (**b**), hydrochar prepared at 200 °C at 500× magnification (**c**), hydrochar prepared at 200 °C at 1500× magnification (**d**), hydrochar prepared at 250 °C at 500× magnification (**e**) and hydrochar prepared at 250 °C at 1500× magnification (**f**).

**Figure 5 materials-18-03374-f005:**
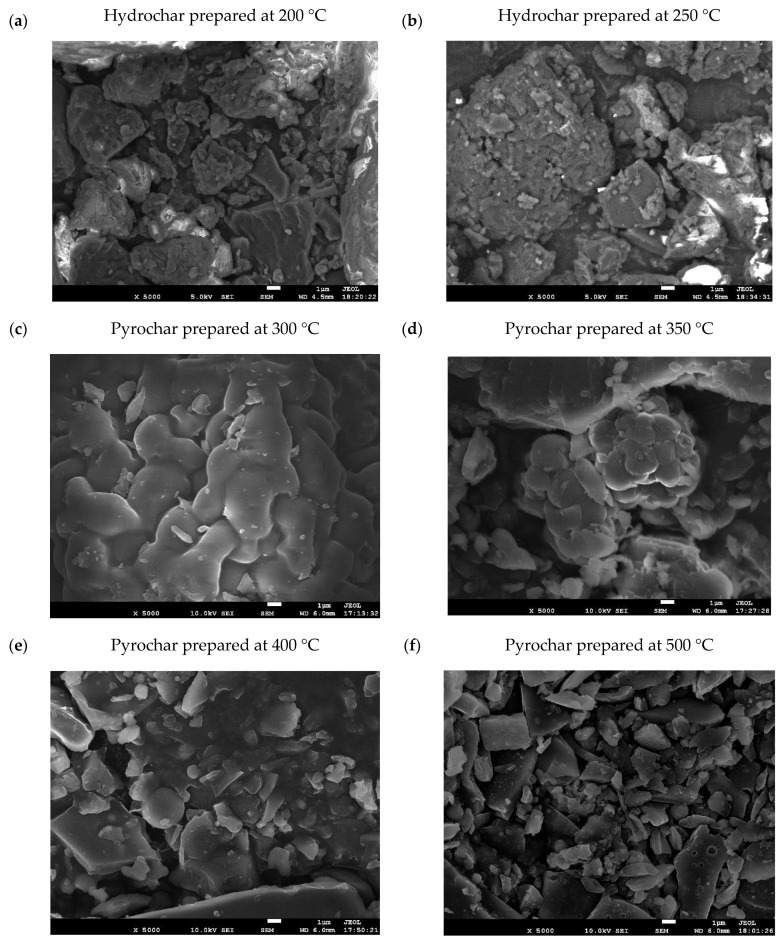
Microstructure of algal cells in hydrochars and pyrochars documented at 5000× magnification.

**Figure 6 materials-18-03374-f006:**
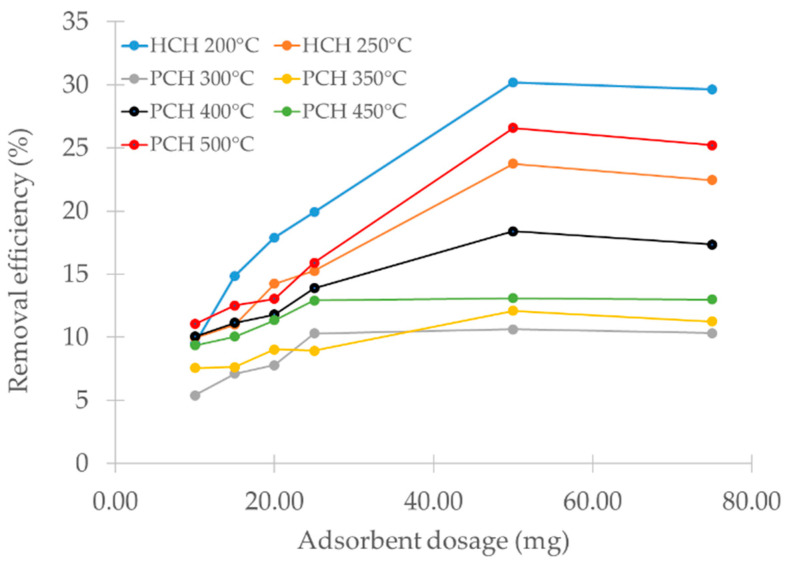
Effect of biochar dosage on the removal of metribuzin by hydrochars (HCHs) and pyrochars (PCHs) prepared under different conditions.

**Figure 7 materials-18-03374-f007:**
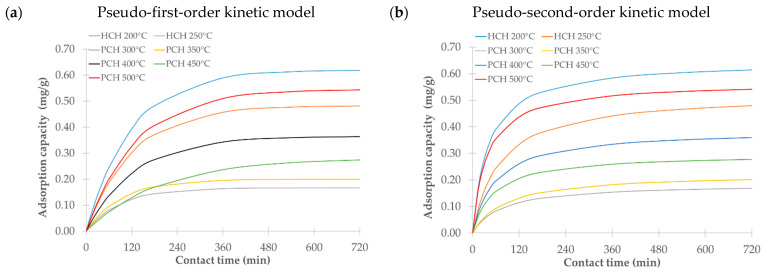
Effect of contact time on metribuzin removal by hydrochars (HCHs) and pyrochars (PCHs) prepared under different conditions: (**a**) fitted to the pseudo-first-order kinetic model; (**b**) fitted to the pseudo-second-order kinetic model.

**Figure 8 materials-18-03374-f008:**
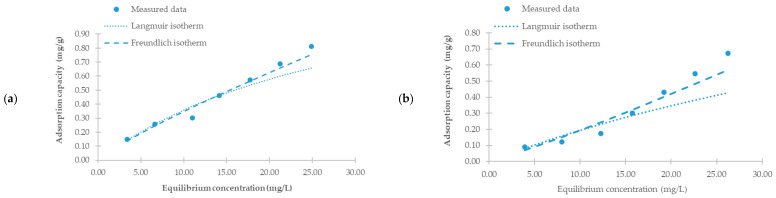
Adsorption isotherms of metribuzin onto hydrochars (HCHs) and pyrochars (PCHs) prepared at various temperatures. Experimental data were fitted using Langmuir and Freundlich isotherm models. Subfigures represent adsorption at (**a**) HCH 200 °C, (**b**) HCH 250 °C, (**c**) PCH 300 °C, (**d**) PCH 350 °C, (**e**) PCH 400 °C, (**f**) PCH 450 °C, and (**g**) PCH 500 °C and (**h**) a comparison of the adsorption capacity modelled for equilibrium concentration 20 mg/L.

**Table 1 materials-18-03374-t001:** Oxidizable organic carbon (C_org_) content in biomass, hydrochars, and pyrochars produced from *Chlorella vulgaris* at various carbonization temperatures. Results are expressed as the mean of three replicates ± standard deviations.

Material	Temperature (°C)	C_org_ (mg/g, Mean ± SD)
Biomass	25	375.13 ± 2.94
Hydrochar	200	384.74 ± 1.11
Hydrochar	250	487.95 ± 2.22
Pyrochar	300	425.13 ± 1.11
Pyrochar	350	477.69 ± 1.92
Pyrochar	400	625.13 ± 2.22
Pyrochar	450	453.33 ± 2.94
Pyrochar	500	528.33 ± 1.11

**Table 2 materials-18-03374-t002:** Specific surface area, BET C constant, and correlation coefficient (r) for pyrochars produced from *Chlorella vulgaris* at 300–500 °C.

Sample	Temperature (°C)	BET Surface Area (m^2^/g)	BET C Constant (–)	Correlation Coefficient (r)
PCH 300	300	0.332	228.745	0.999
PCH 350	350	2.097	6.768	0.999
PCH 400	400	0.688	121.800	0.999
PCH 450	450	3.195	7.297	0.998
PCH 500	500	44.295	112.259	0.999

**Table 3 materials-18-03374-t003:** Kinetic parameters of the pseudo-first-order and pseudo-second-order models for metribuzin adsorption onto hydrochars (HCHs) and pyrochars (PCHs) produced at different temperatures.

Sample	Parameter	Pseudo-First-Order	Pseudo-Second-Order
HCH 200 °C	*k*_1_ (1/min)	6.20 × 10^−1^	
	*k*_2_ (g/mg min)		6.54 × 10^−1^
	*Q*_e_ (mg/g)	6.25 × 10^−3^	3.18 × 10^−2^
	*R* ^2^	0.8984	0.9447
HCH 250 °C	*k*_1_ (1/min)	4.83 × 10^−1^	
	*k*_2_ (g/mg min)		5.35 × 10^−1^
	*Q*_e_ (mg/g)	5.88 × 10^−3^	2.21 × 10^−2^
	*R* ^2^	0.9527	0.9604
PCH 300 °C	*k*_1_ (1/min)	1.85 × 10^−1^	
	*k*_2_ (g/mg min)		1.95 × 10^−1^
	*Q*_e_ (mg/g)	7.36 × 10^−3^	5.57 × 10^−2^
	*R* ^2^	0.9628	0.9528
PCH 350 °C	*k*_1_ (1/min)	2.00 × 10^−1^	
	*k*_2_ (g/mg min)		2.31 × 10^−1^
	*Q*_e_ (mg/g)	1.23 × 10^−2^	4.10 × 10^−2^
	*R* ^2^	0.9301	0.9469
PCH 400 °C	*k*_1_ (1/min)	3.65 × 10^−1^	
	*k*_2_ (g/mg min)		3.94 × 10^−1^
	*Q*_e_ (mg/g)	5.45 × 10^−3^	3.55 × 10^−2^
	*R* ^2^	0.9196	0.9350
PCH 450 °C	*k*_1_ (1/min)	2.81 × 10^−1^	
	*k*_2_ (g/mg min)		3.00 × 10^−1^
	*Q*_e_ (mg/g)	5.16 × 10^−3^	5.12 × 10^−2^
	*R* ^2^	0.9073	0.9373
PCH 500 °C	*k*_1_ (1/min)	5.46 × 10^−1^	
	*k*_2_ (g/mg min)		5.75 × 10^−1^
	*Q*_e_ (mg/g)	7.76 × 10^−3^	3.95 × 10^−2^
	*R* ^2^	0.9015	0.9507

**Table 4 materials-18-03374-t004:** Parameters calculated using the Langmuir and the Freundlich adsorption models.

Sample	Parameter	Langmuir Isotherm	Freundlich Isotherm
HCH 200 °C	*K*_L_ (L/mg)	3.20 × 10^−2^	
	*Q*_max_ (mg/g)	1.48	
	*R* _L_	2.01 × 10^−2^	
	*K*_F_ (mg/g)		4.97 × 10^−2^
	1/*n* (mg/L)		8.45 × 10^−1^
	*R* ^2^	0.9722	0.9884
HCH 250 °C	*K*_L_ (L/mg)	1.39 × 10^−2^	
	*Q*_max_ (mg/g)	1.6	
	*R* _L_	1.86 × 10^−2^	
	*K*_F_ (mg/g)		1.50 × 10^−2^
	1/*n* (mg/L)		1.11
	*R* ^2^	0.9572	0.9807
PCH 300 °C	*K*_L_ (L/mg)	−3.36 × 10^−2^	
	*Q*_max_ (mg/g)	−7.65 × 10^−2^	
	*R* _L_	−6.56 × 10^−1^	
	*K*_F_ (mg/g)		1.11 × 10^−3^
	1/*n* (mg/L)		1.64
	*R* ^2^	−0.5853	0.9841
PCH 350 °C	*K*_L_ (L/mg)	−3.65 × 10^−2^	
	*Q*_max_ (mg/g)	−9.34 × 10^−2^	
	*R* _L_	−4.80 × 10^−1^	
	*K*_F_ (mg/g)		2.54 × 10^−3^
	1/*n* (mg/L)		1.37
	*R* ^2^	0.9683	0.9778
PCH 400 °C	*K*_L_ (L/mg)	7.66 × 10^−2^	
	*Q*_max_ (mg/g)	4.98 × 10^−1^	
	*R* _L_	5.73 × 10^−2^	
	*K*_F_ (mg/g)		4.98 × 10^−2^
	1/*n* (mg/L)		6.10 × 10^−1^
	*R* ^2^	0.9425	0.9747
PCH 450 °C	*K*_L_ (L/mg)	7.32 × 10^−2^	
	*Q*_max_ (mg/g)	3.84 × 10^−1^	
	*R* _L_	7.32 × 10^−2^	
	*K*_F_ (mg/g)		3.82 × 10^−2^
	1/*n* (mg/L)		6.00 × 10^−1^
	*R* ^2^	0.9650	0.9869
PCH 500 °C	*K*_L_ (L/mg)	7.44 × 10^−2^	
	*Q*_max_ (mg/g)	8.16 × 10^−1^	
	*R* _L_	3.58 × 10^−2^	
	*K*_F_ (mg/g)		7.37 × 10^−2^
	1/*n* (mg/L)		6.48 × 10^−1^
	*R* ^2^	0.9793	0.9936

## Data Availability

The original contributions presented in this study are included in the article. Further inquiries can be directed at the corresponding author.

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
