# Peer review of "Chlorella vulgaris-Derived Biochars for Metribuzin Removal: Influence of Thermal Processing Pathways on Sorption Properties"

_materials, 2025, doi:10.3390/ma18143374_

Round 1
Reviewer 1 Report
Comments and Suggestions for Authors
ID: materials-3737292
Title: Chlorella vulgaris-Derived Biochars for Metribuzin Removal: Influence of Thermal Processing Pathways on Sorption Properties
Overview:
Authors report on biochars derived from Chlorella vulgaris via hydrothermal and pyrolytic processes. They present pH, FTIR, SEM, BET and sorption studies. The manuscript is well-written; however, novelty of the study should be emphasized and FTIR discussion should be revised for a comprehensive presentation of the results.
Major comments:
1) Overall, microalgal biochars are well-known. Consider focusing on the novelty of current study. In Ref. 23, authors also report on the difference between hydrochar and pyrochar derived from algae.
2) Section 3.7, why don't authors present BET parameters for hydrochar?
3) Table 2, PCH 300-450, why do surface areaa and constants vary within 1-2 orders of magnitude despite relatively similar morphology of the samples (see Fig. 4)? Does C constant has any physical meaning? Why is C decrease accompanied by area increase, and vice versa?
4) Assessment of the elemental composition will be beneficial to show that only C/O/H are present in the materials, as suggested by FTIR not delving into FIR (<500 cm-1) range. Do authors expect any C-N [https://doi.org/10.3390/su16166870] or Si-O [http://dx.doi.org/10.1016/j.vibspec.2012.06.006] contribution to the FTIR spectra?
Minor comments:
5) Line 158, 680 nm wavelength for biomass gain monitoring: could you elaborate on a reasoning behind this wavelength choice? Typically, organics-related pi-pi* transition provides a major peak at ~300-400 nm.
6) In the section 3.3, consider providing the discussion on the distinguishment between active and exchangeble pH for general audience.
7) Fig. 3: is a narrow line at ~2500 cm-1 an actual line or an instrumental feature?
8) Fig. 3: the resolution of HCH-related spectra seem worse than the one of other spectra, while their signal-to-noise ratio is lower. Could you explain why?
9) Fig. 3, biomass spectrum: the line observed at ~2300 cm-1 is not discussed.
10) Fig. 3: lines at 2300-2800 cm-1 seem unconventional for biochars. In [http://dx.doi.org/10.1016/j.vibspec.2012.06.006], authors suggest somewhat similar band is an instrumental feature which is removed after spectra processing: could you comment on that?
11) Line 475, the band at 2500 cm-1 is not weak. Please revise the fragment.
Reviewer 2 Report
Comments and Suggestions for Authors
Dear Authors,
I had the opportunity to review your manuscript titled “Chlorella vulgaris-Derived Biochars for Metribuzin Removal: Influence of Thermal Processing Pathways on Sorption Properties.” Although the use of microalgae-derived biochar is not entirely novel, the specific application presented here contributes meaningfully to the increasingly relevant topic of persistent organic pollutants, such as herbicides, and may attract the interest of a broad scientific audience.
However, before I can consider making a recommendation regarding your manuscript, I kindly request the following improvements. My comments are intended solely to enhance the scientific quality and clarity of your work. If the authors revise the manuscript by adequately addressing the points raised, I will be pleased to re-evaluate the submission.
In particular, the authors should strengthen the discussion of the study's novelty. Please clearly articulate the originality, motivation, and hypothesis of the work at the end of the Introduction section, preferably in the final or penultimate paragraph.
The manuscript would benefit from several clarifications and formatting improvements to enhance its clarity, consistency, and scientific rigor. For example, in lines 162–164, the meaning of “A PPL100” is not immediately clear. If this refers to a specific reactor or piece of laboratory equipment, its full name and function should be clearly explained upon first mention. Additionally, in line 343 and other sections of the manuscript, the citation format “according to [35]” appears somewhat unorthodox. Rather than citing numerically without context, the authors might consider integrating author names into the sentence structure for smoother readability. This stylistic issue should be reviewed throughout the manuscript for consistency.
Within the FTIR discussion (L462, 469, and 474), the formatting appears more suitable for book chapters than journal articles, as it employs multiple subheadings within a single results section. This structure interrupts the narrative flow and should be revised into unified, continuous-text paragraphs to align with standard journal formatting. Moreover, the FTIR analysis presented in Figure 3 appears to lack baseline correction, which may affect the quality and interpretability of the spectral data. The authors are encouraged to apply baseline correction, either through manual adjustment using software like Origin or via automated computational methods. Doing so would significantly improve the visual clarity and analytical value of the FTIR plots.
A minor but important issue occurs in line 624, where the chemical name “metribuzine” appears. The authors should ensure consistent use of terminology, choosing either “metribuzin” or “metribuzine” and applying it uniformly throughout the text.
In terms of content, the abstract effectively identifies a research gap by stating that paired evaluations of hydrochar and pyrochar from a single feedstock are rare, especially regarding pesticide adsorption. However, the abstract does not sufficiently explain why this gap matters. The authors could strengthen this section by clarifying that using a single feedstock enables more accurate comparisons between thermal conversion pathways, isolating processing variables from feedstock-related differences. Furthermore, the claim of novelty based on the absence of chemical activation should be accompanied by a brief mention of its practical advantages, namely, simplified production, lower costs, and reduced chemical waste, thereby reinforcing the study’s relevance to sustainable water treatment practices.
The introduction is comprehensive in outlining the environmental risks of pesticide contamination, yet it presents metribuzin as a target compound only after discussing general pesticide concerns. Introducing metribuzin earlier and explicitly linking its physicochemical properties—such as its polarity, solubility, and persistence—to its behavior in aquatic systems would create a more focused and logical progression. Similarly, the discussion of Chlorella vulgaris as a biochar feedstock notes its high nitrogen content and homogeneous composition but does not fully explain how these characteristics could enhance metribuzin adsorption. A deeper explanation of how nitrogen-rich functional groups could interact with nitrogenous pesticides through π–π interactions, hydrogen bonding, or electrostatic attraction would bolster the rationale for choosing this microalgae species.
In the materials and methods section, further precision is required in several areas. For instance, the drying process of the algal biomass is described as lasting “a maximum of 24 hours” at 105 °C. This time-based criterion may result in variable residual moisture content. A more robust approach would involve drying until a constant weight is achieved, ensuring comparability across samples. Similarly, descriptions of cooling steps after thermal treatments, phrased as “cooled to laboratory temperature”, are vague. Since cooling rate can influence structural and chemical properties of carbonized materials, more specific details (e.g., passive air cooling, specific time or rate) would improve methodological transparency.
In the BET surface area analysis, the degassing temperature of 200 °C for 72 hours might be insufficient to fully remove adsorbed species from pyrochars produced at higher temperatures, potentially underestimating surface area. Although the authors acknowledge this limitation in the discussion, it remains a critical point. For the batch adsorption experiments, specifying “laboratory temperature” is also insufficiently precise; a defined value (e.g. 25 ± 1 °C) should be provided, as adsorption is temperature-sensitive. Moreover, while the authors chose not to include error bars in graphical data due to low standard deviations, they should still report the average or range of standard deviations in the text. An even better approach would be to display data replicates as individual points or include a supplementary table listing these values for transparency.
The ash content section mentions the potential impact of high ash levels on biochar properties but does not explicitly discuss how ash content in this study’s biomass influenced surface area, functional groups, or metribuzin removal performance. A more detailed analysis would improve this section. In the pH discussion, observed trends are attributed to functional group behavior and mineral content, but without directly linking these mechanisms to evidence from FTIR or ash composition. Integrating these datasets would strengthen the argument.
The FTIR analysis is thorough in describing functional group transformations, but the section largely stops at description. To enhance its scientific value, the discussion should explicitly connect how variations in functional groups, such as a higher prevalence of hydroxyl or carboxyl moieties in hydrochars, relate to increased chemisorption of metribuzin. The authors correctly note that HCH 200 °C exhibited the best performance but fail to deeply explore how this correlates with the structural features identified. The SEM results are well articulated in terms of morphology, but they would benefit from quantitative correlation with adsorption properties. The moderate porosity of hydrochar at 200 °C, for instance, is not supported by BET data, since BET analysis was only applied to pyrochars. Either provide qualitative estimates from SEM or acknowledge this limitation explicitly.
The discussion of BET results reveals that PCH 500 °C had the highest surface area, yet HCH 200 °C had superior adsorption capacity. This apparent contradiction is not adequately addressed. The authors should emphasize that for polar compounds like metribuzin, surface chemistry (via chemisorption) may play a more critical role than surface area, especially given the kinetic data supporting pseudo-second-order adsorption behavior. This insight should be more strongly integrated into the interpretation.
Regarding biomass dose, the decline in adsorption efficiency at 75 mg is attributed to particle agglomeration. While this is plausible, no empirical evidence is offered. If particle size analysis or visual aggregation was not performed, the authors should frame this explanation as a hypothesis and suggest it for future validation. Similarly, the kinetic modeling shows that most samples follow pseudo-second-order behavior, implying chemisorption, yet the specific interactions involved, such as hydrogen bonding or electron exchange, are not discussed. Furthermore, the unique pseudo-first-order fit of HCH 250 °C is noted but not explored in terms of its functional group composition or morphology.
The adsorption isotherms section correctly notes that the Freundlich model best fits most data, indicating surface heterogeneity. This finding should be expanded by linking the observed heterogeneity to the functional and structural features identified through FTIR and SEM. The Langmuir model’s poor performance, particularly for PCH 300 °C and 350 °C, which produced negative Qmax and KL values, is a critical result. The authors mention this briefly but should emphasize it more strongly as evidence against monolayer adsorption. Additionally, statements that suggest “irreversibility” based on RL values alone should be tempered unless supported by desorption experiments. A more cautious phrasing, such as “strong binding affinity,” would be more appropriate.
Across the results sections, greater integration of findings is needed. For instance, retention of functional groups (FTIR), intact morphology (SEM), and adsorption capacity (kinetics) for HCH 200 °C should be directly tied together to offer a coherent mechanistic explanation. The manuscript would also benefit from proposing concrete hypotheses about the dominant adsorption mechanisms (such as π–π stacking, hydrogen bonding, or electrostatic interactions) between metribuzin and biochar surface groups. Where quantitative porosity data is lacking, especially for hydrochars, this limitation should be acknowledged, and future work using CO₂ adsorption or complementary techniques should be proposed.
In the conclusions, the authors correctly state that HCH 200 °C exhibited superior adsorption due to retained morphology, functional group richness, and moderate porosity. However, since hydrochar porosity was not quantified in this study, the “moderate porosity” claim lacks direct support. Each conclusion should ideally be referenced back to specific data or figures—for instance, morphology confirmed by SEM (Figure 4b) and functional group richness from FTIR (Figure 3). Additionally, the concluding remarks on the potential of Chlorella vulgaris for nutrient recovery and CO₂ capture, while relevant to the broader context of circular bioeconomy, are not directly derived from this study and should be reframed as broader implications rather than results.
Finally, when comparing adsorption capacities with those of chemically activated biochars from the literature, the authors should highlight the trade-offs. Emphasize that while peak capacity might be lower, unmodified biochars like those produced here offer advantages in terms of cost, simplicity, and environmental sustainability, making them suitable for decentralized, low-tech water treatment solutions. Suggesting future studies on desorption behavior, advanced surface characterization, and computational modeling of adsorbate adsorbent interactions would provide valuable next steps and elevate the scientific contribution of the work
Reviewer 3 Report
Comments and Suggestions for Authors
Dear Authors,
This study investigates the comparative performance of hydrochar and pyrochar derived from Chlorella vulgaris for the adsorption of the herbicide metribuzin from aqueous solutions. The work is scientifically sound, well-structured, and combines comprehensive material characterization with adsorption kinetics and isotherm modeling, offering valuable insight into the influence of thermal processing on sorbent properties. I recommend accepting the paper after minor revision, with specific comments aimed at improving clarity, consistency in data presentation, and strengthening the discussion on adsorption mechanisms.
Specific comments:
Introduction
I suggest adding some data on the detected levels of metribuzin in water matrices
The aims/hypotheses/research question is not well defined.
Line 68: "biochar is the most promising." This should be explained better, most promising in terms of what? Sorption capacity, sustainability, economic benefits...?
Materials and Methods
Section 2.4: Were pH measurements performed in replicates?
Section 2.5: Was pH controlled or monitored during the batch adsorption experiment, as metribuzin speciation may vary? Were there any control samples included to exclude loss of analyte through processes other than sorption to the sorbent? Is the concentration range in the experiment environmentally relevant?
Section 2.7: Mention the column type and dimensions for HPLC analysis to ensure reproducibility.
Results
Adsorption capacities in Figure 8 are presented in mg/g, and Qmax in Table 4 are given in mg/kg; however, it seems one of these is incorrect, as this way the Qmax values are 1000 times lower than the equilibrium adsorption capacities. Please check the units. Also, if the units in Table 4 are correct, Qmax values <1 mg/kg are quite low. Discuss whether this is expected for metribuzin and observed in literature with other carbonaceous sorbents as well.
Where means of replicate measurements are given (in figures and tables), error bars or confidence intervals should be shown.
Round 2
Reviewer 1 Report
Comments and Suggestions for Authors
ID: materials-3737292
Title: Chlorella vulgaris-Derived Biochars for Metribuzin Removal: Influence of Thermal Processing Pathways on Sorption Properties
Authors have addressed my comments satisfactory. However, some wording and presentation issues regarding the revised text. I recommend a minor revision of the manuscript.
1) Lines 201-203, "Once it reached this temperature, the mixture was opened and centrifuged (Nahita model 2640/12) to separate the solid and liquid phases.": denoting centrifuge model and centrifugation parameters will be beneficial.
2) Lines 528-529, "lower aromatic content, which reduce the peak sharpness and signal intensity": please elaborate on the interplay betwee the aromaticity, peak width and intensity. Non-aromatic conjugated polymers, such as polyacetylene, possess well-resolved sharp lines [10.3390/coatings14091216].
3) Lines 579-580, "FTIR spectra show a gradual loss of hydroxyl (3600–3200 cm⁻¹), carboxyl (1745–1700 cm⁻¹), and amide groups (around 1655 cm⁻¹) with increasing treatment temperature.": first, the loss isn't gradual, as hydroxyl peaks are present on one sample only, carboxyl peaks possess similar intensity for 450 and 500; second, these changes are not comprehensible, as the lines overlap each other. Consider re-normalizing them, show the lines with an offset and denoting the peak positions for better visibility.
Reviewer 2 Report
Comments and Suggestions for Authors
The article has been improved and is now ready to be considered for publication. Please revise the conclusion to reduce the number of paragraphs, as it currently has too many (ideally, it should be a single paragraph).
